# Prevalence of non-tuberculous mycobacteria among people with acid-fast positive presumptive tuberculosis in Mali

**Aissata Boubakar Cisse**[1]*, **Anna S. Dean**[2], **Armand Van Deun**[3†], **Jelle Keysers**[3], **Willem-Bram De Rijk**[3], **Mourad Gumusboga**[3], **Hawa Samake**[1], **Seydou Arama**[1], **Bassirou Diarra**[5], **Ibrahim Djilla**[1], **Fatoumata N. Coulibaly**[1], **Hawa Simpara**[1], **Mamadou Berthe**[1], **Khadidia Ouattara**[6], **Yacouba Toloba**[6], **Ibrehima Guindo**[1], **Bouke de Jong**[3], **Leen Rigouts**[3,4]

1 National Institute of Public Health, Bamako, Mali, 2 Word Health Organization, Geneva, Switzerland, 3 Mycobacteriology Unit, Institute of Tropical Medicine, Antwerp, Belgium, 4 Belgian Coordinated Collections of Microorganisms, Antwerp, Belgium, 5 University Clinical Research Center, Bamako, Mali, 6 University Teaching hospital of Point G, Bamako, Mali

† Deceased.
* aissatacisse@yahoo.fr

**Data Availability Statement:** All relevant data are within the manuscript and its Supporting Information files.

## Abstract

### Background

Non-tuberculous mycobacteria (NTM) are environmental agents that can cause opportunistic pulmonary disease in humans and animals, often misdiagnosed as tuberculosis (TB). In this study, we describe the cases of NTM identified during the first national anti-TB drug resistance survey conducted in Mali and explore associated risk factors.

### Methods

Sputum was collected from people presenting for pulmonary TB diagnosis from April to December 2019, regardless of age. Microscopy-positive patients were enrolled and tested using the Xpert MTB/RIF assay. A patient who tested negative for the *Mycobacterium tuberculosis* complex (MTBC) was tested for the presence of mycobacteria by amplification of the IS*6110* and 16SrRNA (16S) genes through double quantitative real-time PCR, followed by nested PCR and Sanger sequencing of the IS*6110*-negative samples for NTM species identification.

### Results

1,418 sputum smear-positive patients were enrolled, including 1,199 new cases, 211 previously treated cases, and 8 whose previous treatment history was unknown. Based on the results of Xpert MTB/RIF assay and in-house PCR methods, 1,331 (93.9%) patients were positive for MTBC, 48 (3.4%) for NTM, and no species identification was possible for 39 (2.7%). Advanced age of 65 and over with an OR 8.8 (95% CI 2.3–33.2 and p = 0.001) and previous TB treatment with an OR 3.4 (95% CI 1.2–9.6 and p = 0.016) were the risk factors

**Funding:** 1st tuberculosis Drug resistance survey financed by the Global Fund.

**Competing interests:** The authors have declared that no competing interests exist.

statistically associated with NTM detection. *M. avium complex* (MAC) was the predominant NTM species, detected in 20 cases.

## Conclusion

Detection of NTM in people presumed to have TB is an ongoing challenge, confounding correct TB diagnosis. Concomitant use of microscopy and GeneXpert testing among at-risk individuals could improve patient management.

## 1. Introduction

The *Mycobacterium* genus groups species of the *Mycobacterium tuberculosis* complex (MTBC) responsible for tuberculosis (TB) in humans and animals, *M. leprae*, the causative agent of leprosy, and the species of mycobacteria commonly known as atypical or non-tuberculous mycobacteria (NTM), including *M. ulcerans* responsible for Buruli ulcer. Unlike members of the MTBC group, NTM are not obligatory parasites of humans, yet normal residents of soil and water, and can be found in natural and treated water sources [1]. More than 200 species of NTM have been officially recognised [2], of which around 25 are known to be strongly associated with disease in humans. Some species have been associated with pulmonary diseases causing symptoms similar to TB [1]. Because of their habitat, humans are exposed to these bacteria daily. Therefore, NTM disease must be distinguished from simple colonisation or contamination of clinical samples, for example from tap water [1, 3]. Unlike TB, the global epidemiology of disease caused by NTM is not well known. NTM isolation from clinical specimens is mainly reported in industrialised countries, with variable prevalence and incidence. Studies based on pulmonary specimen isolates reported a prevalence of 1.4 to 6.6 per 100,000 persons in the USA from 2004 to 2006 [4], 9.8 in 2010 in Ontario, Canada [5], and 5.8 per 100,000 in Germany [6] in 2020. Incidence of 6.1 per 100,000 in 2012 in England [7] and 5.3 per 100, 000 in Germany in 2020 [6] have also reported. In TB-endemic countries, NTM is being reported less frequently and mainly in populations at risk, particularly people with predisposing conditions or compromised immunity [8]. However, experiences in industrialised countries have shown that a declining TB burden has also increased the number of NTM cases detected. With the strengthening of TB programmes in another setting, we may also anticipate a similar scenario, posing a growing diagnosis and clinical challenge in low-and middle-income countries [9]. The diagnosis of NTM lung disease is based on clinical, radiological, and microbiological criteria [1]. Laboratory-based detection of NTM, with differentiation from MTBC and identification to the species level, is mainly inaccessible in most resource-limited countries. Microscopy, the most accessible technique, identifies MTBC and NTM as acid-fast bacilli (AFB) but fails to distinguish them. Since 2010, the World Health Organization (WHO) has recommended rapid molecular tests such as GeneXpert MTB/RIF (Xpert) as the initial test for TB diagnosis, exhibiting a higher sensitivity and specificity [10]. This assay only identifies the presence of MTBC species in the sample. A negative result for MTBC by Xpert obtained on AFB-positive sputum could indicate NTM [11]. In Mali, cases of NTM infection have been described, particularly in patients under investigation for failure of anti-TB treatment or who have relapsed after TB cure [12]. Following the introduction of Xpert in the country, plausible cases of NTM infection with an AFB-positive smear and a negative Xpert have been reported more frequently [13].

In this study, we describe the proportion of MTBC and NTM detected in patients presenting for initial or previously treated TB diagnoses in Mali in the context of a nationwide drug resistance survey [14]. We also explored risk factors associated with NTM detection.

## 2. Methods

### 2.1. Design of the study

From April to December 2019, consecutive eligible patients were recruited from 78 of 86 TB diagnostic laboratories, representing 91% of the country's TB laboratory network. Eight laboratories notified less than 10 new bacteriologically confirmed pulmonary TB cases in 2017 and were not included for logistic reasons. Eligible participants were defined as people undergoing evaluation for pulmonary TB, regardless of age, with and without a prior treatment history. The study aims to measure the percentage of new pulmonary TB cases resistant to rifampicin. To achieve this, 994 new cases will be sampled over a 4-month recruitment period [15].

### 2.2. Sample collection and patient enrolment

Two sputum samples of approximately 5 mL were collected from each patient in the participating laboratories. Smears were made directly from freshly collected sputum, stained according to the acid-fast staining technique used at the center (auramine for most sites and Ziehl-Neelsen for several small centers), and in line with international guidelines for diagnosing TB by microscopy [16]. If one of the two sputum samples tested positive for AFB, the patient was included in the drug resistance survey after providing informed consent, and an additional sample was collected.

The nurse completed the study questionnaire with the recruited patient. The questionnaire covered socio-demographic data, information on previous TB treatment, and pulmonary disease history (duration and chronicity of respiratory signs). The patient also given important clinical and treatment-related information. All enrolled participants received anti-TB treatment. New patients were started on first-line TB treatment based on AFB microscopy.

According to national guidelines, treatments were initiated based on the Xpert MTB/RIF testing results of patients with previously treated TB.

### 2.3. Initial GeneXpert testing and specimen storage

The first two sputum samples collected were preserved in 5 mL of OMNIgene® SPUTUM (OM-S) decontamination reagent (DNA Genotek Inc, Canada) according to the manufacturer's recommendations [17]. On the over hand, the third sample was stored in 10 mL of 96% ethanol. The conditioned samples were transported by public transport at ambient temperature in triple packaging to the National TB Reference Laboratory (NTRL) in Bamako.

The Xpert MTB/RIF assay (Cepheid, CA, USA) was performed at the NTRL using 2 mL of one OM-S-preserved sample, as per the manufacturer's recommendations (Cepheid and Genotek).

Leftovers of the OM-S-preserved samples were then stored in cryotubes as sediment obtained after centrifugation at 4000 g for 15 minutes and mixed with 96% alcohol at an equal volume (0.5 mL sediment and 0.5 mL ethanol).

### 2.4. Molecular detection and identification of NTM species

According to the study protocol, samples from the following patients were transported at ambient temperature to the WHO Supranational TB Reference Laboratory at the Institute of

Table 1. Primers and probes for qPCR to simultaneously detect *Mycobacterium tuberculosis* complex and all mycobacterial species.

| Target | Name | Sequences |
|---|---|---|
| IS*6110* gene | IS*6110*_S<br>IS*6110*_AS | 5'-CCTGAAAGACGTTATCCACCATAC-3'<br>5'-GGC-TAG-TGC-ATT-GTC-ATA-GGA-G-3' |
| | IS6110_P_DFO | DFO-TCT-CAG-TA(lnC)-AC(lnA)-TC(lnG)-ATC-CGG-T-BHQ2 |
| 16S gene | Myco_Genus_S_v3<br>Myco_Genus_AS_v2 | 5'-ATGCAAGTCGAACGGAAAGG-3'<br>5'-TCTGCCCGTATCGCC-C-3' |
| | Myco_Genus_P_v2 | FAM-CAA-AC(lnC)-AC(lnC)-TA(lnC)-GA(lnG)-CTC-EDQ |

Tropical Medicine (ITM) in Antwerp, Belgium, for molecular detection and identification of NTM species:

- cases with rifampicin resistance by Xpert MTB/RIF, regardless of anti-TB treatment history;

- all previously treated cases, regardless of Xpert MTB/RIF test result;

- and cases for which Xpert did not detected MTBC.

In this article, we describe only the molecular methods employed and the results obtained for species identification of probable NTM.

Upon arrival at ITM, ethanol-preserved samples were centrifuged at 4000 g for 15 minutes, digested by overnight proteinase K incubation, followed by lysis using an internal lysis buffer, after which DNA was extracted using the Modified Maxwell-LEV® automated extraction method (Promega, USA) [18]. A duplex quantitative real-time polymerase chain reaction (qPCR) was performed, targeting the IS*6110* gene to detect the MTBC and the 16S gene to detect all species of the *Mycobacterium* genus. The primers and probes used are presented in Table 1, while the interpretation of results is depicted in Table 2.

Subsequently, a nested conventional PCR targeting the 16S gene was performed on samples identified as NTM by the duplex qPCR. The primers and probes used were as follows:

- P1 (first PCR): 5'-TGCTTAACACATGCAAGTCG-3'

- P2 new (first PCR): 5'-TCTCTAGACGCGTCCTGTGC-3'

- P7 (nested PCR): 5'-CATGCAAGTCGAACGGAAAGG-3'

- P16 new (nested PCR): 5'-AAGCCGTGAGATTTCACGAACA-3'

Amplicons that were positive on the agarose gel underwent Sanger sequencing. A negative PCR result was interpreted as the absence of MTBC or NTM.

Sequencing was subcontracted to BaseClear (BE Leiden, The Netherlands). After entering the sequence into the Basic Local Alignment Search Tool (BLAST) of the National Centre for

Table 2. Interpretation of duplex IS*6110*-16S qPCR results.

| 16S qPCR Cycle threshold (Ct) | IS*6110* qPCR result | |
|---|---|---|
| | Positive | Negative |
| **Positive with any Ct value** | - | No MTBC |
| **Positive and ≥IS*6110* Ct** | MTBC | |
| **Positive and <IS*6110* Ct** | MTBC + NTM | |
| **Negative** | MTBC<br>(low bacillary load) | No mycobacteria species identification possible |

MTBC = *Mycobacterium tuberculosis* complex; NTM = non-tuberculous mycobacteria: Ct = Cycle threshold

Biotechnology Information (NCBI), we compared it to reference strains sequences on NCBI such as ATCC 25791, ATCC BAA-614, ATCC 27282, ATCC 51457, 4773, ATCC 700504, ATCC 25276, ATCC 35752, ATCC 13950. We interpreted the results obtained as follows:

- If a sequence match was found on the NCBI, the NTM species was identified.

    ○ In case of 100% similarity (no mismatches), the species name was assigned.

    ○ In case 1–3 bp mismatches with the closest species, the "species-like" name was assigned.

- If no match or >3 bp difference with the closest species was found on NCBI, the result was interpreted as *Mycobacterium* species.

- If there were overlapping peaks in the sequence, indicating the presence of several NTM, the result was interpreted as "Mixture of NTM ".

## 2.5. Data analysis

The data collected using the survey and sample submission forms were entered into EpiInfo version 7.2.5.0. The variables analysed were age, sex, region of residence, history of illness, and history of anti-TB treatment. Percentages, medians, and means were calculated by crosstab, and p-values and odds ratios with 95% confidence intervals were generated using regression analysis in IBM SPSS Statistics version 21X64 (IBM Corp., NY, USA).

## 2.6. Ethical considerations

The drug resistance survey protocol in which a secondary analysis of samples is described in the informed consent was accepted by the National Institute of Public Health Research ethics committee by decision No. 08/2018/CE-INRSP on March 21, 2018. The study objectives and expected results were clearly explained in a language understandable to the patient. Patients were asked permission to use their samples for additional analyses, including outside Mali. We obtained oral and written consent from patients 18 years of age or older. For participants under 18, their consent to take part in the study was obtained in the presence of a witness other than the parent or companion, after receiving prior oral and written permission from the parent or guardian.

## 3. Results

### 3.1. Mycobacteria detected

A total of 1,418 AFB-positive people with presumed pulmonary TB were enrolled in the drug resistance survey, including 1,199 new cases, 211 previously treated cases, and 8 whose previous treatment history was unknown (Fig 1). Based on Xpert MTB/RIF results, 1,296 (91.4%) patients were positive for MTBC, while 1MTBC was not detected for 22 (8.6%). The duplex IS6110-16S qPCR performed for 119 of these 122 patients identified another 35 MTBC, bringing the total number of patients with MTBC to 1,331 (93.9%). Of these 35 additional cases, MTBC alone was detected for six, while 29 carried probable mixtures of MTBC and NTM.

For 81 (68,1%) of 119 patients tested, only the 16S target reacted positively (presumed NTM), while the PCR remained negative for both targets in 3 (2,5%) patients. The nested conventional 16S PCR performed for the 81 presumed NTM detected by IS*6110*-16S qPCR was positive for 48 or 3.4% (48/1418) of all enrolled patients, of which 33 could be identified to the species level by sequencing. The remaining 33 of 81 presumed NTM had a negative nested PCR result, thus no sequencing could be performed (Fig 1).

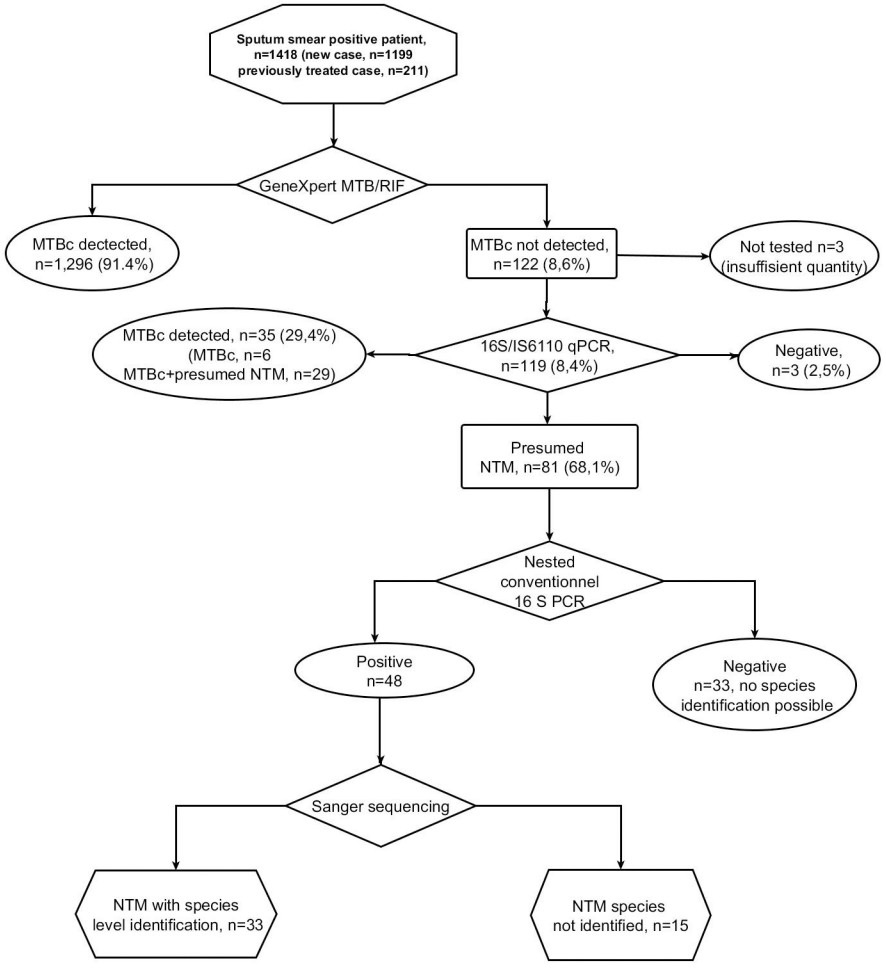

**Fig 1. Sample and analysis flow with summary of results.** MTBC = *Mycobacterium tuberculosis* complex; n = number; NTM = non-tuberculous mycobacteria.

Eleven species or groups of species were identified. The *M. avium* complex (MAC) was the most common, representing 20 cases or 42% of all species detected. The *M. intracellulare* complex was found in 13 patients (27%), *M. avium* in five patients (10%), and *M. colombiense* in two patients (Fig 2).

## 3.2. Detection of NTM in relation to smear-microscopy results

The microscopy laboratories provided AFB quantification result for 1414 of the 1418 patients.

A significantly higher proportion of NTM was detected in patients with lower AFB-grades: 7% for 1+ and scanty (Table 3), compared to 2% among 3+ patients (respectively OR 4.065, 95% CI: 1.876–8.808, p = 0.000 and OR 4.3, 95% CI: 2.012–9.189, p = 0.000). The difference was not significant compared to grade 2+ samples (p = 0.462).

## 3.3. Socio-demographic, clinical characteristics and risk factors for NTM infection

Male patients were more common among study participants and among those with identified NTM, with a sex ratio of respectively 2.1 and 1.5 when compared to women, yet NTM

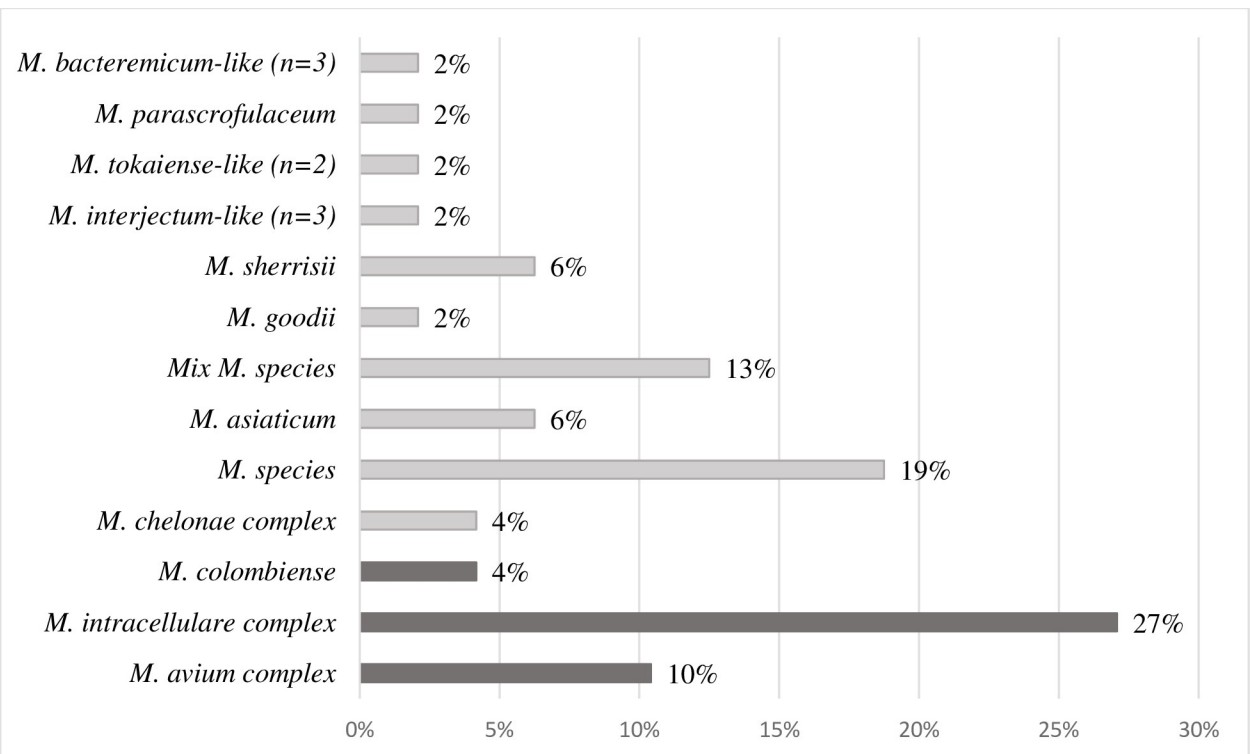

**Fig 2. Sequencing results for the identification of NTM species.**

presence was not associated with gender (Table 4). On the contrary, older age was significantly associated with NTM detection. The median age was 37 years for all patients and 59.5 years for those with NTM detected. Compared to the under-34 age group, the 35–64 age group had greater odds (OR 3.4, 95% CI: 1.1–10.3, p = 0.03) of NTM detection, which further increased for the over-65 age group (OR 8.8, 95% CI: 2.3–33.2 p = 0.001 (Table 4). Also, previously treated patients were more likely than new cases to have NTM in their sputum (OR 3.4, 95% CI: 1.2–9.6, p = 0.016) (Table 4). We found no significant associations between NTM detection and HIV status or chronic respiratory signs. Regarding the patient's residence, patients from the Mopti region were more likely to have NTM detected than Bamako residents (OR 4.1, 95% CI: 1.0–16.4, p = 0.04). (Table 4).

**Table 3. Proportion of NTM and MTB detection in relation to quantitative sputum-smear microscopy results.**

| Final sputum smear result | MTBC n (%) | NTM n (%) | No species identification possible n (%) | Total n (%) |
|---|---|---|---|---|
| Scanty | 167 (83) | 13 (7) | 21 (10) | 201 |
| 1+ | 170 (89) | 14 (7) | 8 (4) | 192 |
| 2+ | 259 (94) | 7 (3) | 8 (3) | 274 |
| 3+ | 731 (98) | 14 (2) | 2 (0) | 747 |
| Not quantified | 4 (100) | 0 | 0 | 4 |
| **Total** | **1331 (93.9)** | **48 (3.4)** | **39 (2.7)** | **1418 (100)** |

MTBC = *M. tuberculosis* complex; NTM = non-tuberculous mycobacteria; n = number

**Table 4. Multivariate logistic regression analyses of demographic and clinical variables as risk factors of NTM infection.**

| Variables | | Total patients n (%) | NTM detected n (%) | OR | p-value |
|---|---|---|---|---|---|
| **Gender n = 1379** | Female | 434 (31.5) | 19 (39.6%) | 0.8 [0.3–2.04] | 0.7 |
| | Male | 945 (68.5) | 29 (60.4%) | | |
| **Age n = 1379** | Median | 37 | **59,5** | | |
| **Age group** | 0–34 years (Reference) | 603 (43.7) | 4 (8.3) | | |
| | 35–64 years | 659 (47.8) | 29 (60.4) | **3.4 [1.1–10.3]** | **0.03** |
| | ≥65 years | 117 (8.5) | 15 (31.3) | **8.8 [2.3–33.2]** | **0.001** |
| **History of treatment n = 1371** | New | 1165 (84.5) | 30 (62.5) | | |
| | Previously treatment | 206 (14.9) | 18 (37.5) | **3.4 [1.2–9.6]** | **0.016** |
| **HIV status n = 1092** | Positive | 77 (5.6) | 4 (8.3) | 2.8 [0.8–8.9] | 0.07 |
| | Negative | 1015 (73.6) | 29 (60.4) | | |
| **Chronic clinical signs n = 1174** | Yes | 442 (37.6) | 24 (60) | 0.7 [0.3–1.8] | 0.5 |
| | No | 752 (62.4) | 16 (40) | | |
| **Region of residence at the time of diagnosis n = 1348** | Bamako (Reference) | 451 (33.5) | 5 (10.9) | | |
| | Kayes | 101 (7.5) | 4 (8.7) | 2.7 [0.5–12.7] | 0.2 |
| | Koulikoro | 217 (16.1) | 7 (15.2) | 1.5 [0.3–7.4] | 0.5 |
| | Sikasso | 155 (11.5) | 12 (26.1) | 2.9 [0.7–10.8] | 0.10 |
| | Ségou | 147 (10.9) | 7 (15.2) | 3.2 [0.8–12.7] | 0.09 |
| | Mopti | 191 (14.2) | 7 (15.2) | **4.1 [1.02–16.4]** | **0.04** |
| | Northern regions | 86 (6.4) | 4 (8.7) | 4.2 [0.7–25.2] | 0.1 |

NTM = non-tuberculous mycobacteria; n = number; OR = odds ratio

## 4. Discussion

The proportion of sputum-AFB-positive patients and MTBC detected by Xpert MTB/RIF among people presumed to have TB in our study was slightly lower than the rate reported in other drug resistance surveys, ranging from 95.8%-97.7% [19–22]. The confirmed proportion with NTM in our study was higher than in TB drug-resistance surveys in Pakistan (0.7%, 13/1972) [19] and in India (1.2%, 35/2,938) [23], yet lower than the 6.4% (317/4917) observed in China [24]. This result may reflect regional differences in the occurrence of these mycobacteria and the diseases they cause worldwide [25, 26].

NTM detection was more likely among paucibacillary patients as determined by smear microscopy, corroborating findings from Nigeria where the AFB-smear grading was relatively lower for patients with NTM detected than for those with MTBC [27], and higher AFB grading was predictive for MTBC detection [28]. With age, the risk of developing NTM infection increases. Still, we did not find a significant association between gender and NTM infection; contrary to other studies, females have been more frequently associated with NTM infections, especially at older ages [11, 29, 30]. Previous anti-TB treatment was another risk factor for NTM infection in our study, confirming findings from NTM pulmonary disease in China and South Africa [31, 32]. Nevertheless, we did not find an association between NTM detection and chronic respiratory signs in our study, in contrast with the described association between NTM infection and lung disease or chronic respiratory symptoms in studies from China and Denmark [33, 34].

HIV infection has frequently been identified as a risk factor for NTM infections and disease, especially in situations of advanced immunosuppression [27, 35, 36]. In our study, the trend toward an association between HIV positivity and NTM detection was not significant,

corroborating previous observations in Mali [37]. No information about anti-retroviral treatment was gathered as part of the drug resistance survey. Still, such treatment is to be started upon HIV diagnosis as per national guidelines in Mali. While patients from the Mopti region had higher odds of having NTM detected in their sputum than those from the Bamako region, further investigation of the environmental conditions in these regions would be needed, especially as NTM are environmental bacteria and have often been isolated at different frequencies in different geographical areas [25, 30, 38]. MAC species were the most commonly NTM detected in our survey. The predominance of MAC has been reported in pulmonary NTM infections in many countries around the world [25, 26, 39, 40]. In Mali, between 2004 and 2013, *M. avium* was the most frequently identified species of the MAC complex [12, 37].

Employing an additional in-house qPCR for IS*6110* in our study has increased the detection of MTBC. The MTBC-specific repetitive element IS*6110* is a suitable target for MTBC detection assays [41]. The applied qPCR IS*6110* has shown good sensitivity and specificity for detecting MTBC compared to culture [42] and higher performance when compared to the Xpert MTB/RIF assay [43]. The observed high Ct values in our study indicate, however, a low quantity of MTB DNA in samples from these patients, which may have been positive upon repeated Xpert testing [44] or by using the Xpert Ultra test, which is more sensitive by targeting both IS*6110* and another repetitive element IS*1081* for MTBC detection, in addition to the *rpoB* gene for assessing rifampicin susceptibility [45]. The higher positivity rate by qPCR compared to the conventional nested PCR we observed in our study could be partially attributable to the non-specificity of the 16S primers employed for the in-house qPCR, which also detects species belonging to closely related genera like *Rhodococcus* and *Nocardia*, hence potentially overestimating the NTM rate. These closely related species can indeed be detected in smear-positive samples not containing tubercle bacilli, as documented in the USA, with 78.4% (134/171) of AFB-positive specimens yielding a positive culture with NTM or *Nocardia* spp, in comparison 14.6% (25/171) of AFB-positive specimens grew MTBC, and another 7% (12/171) remained negative in culture [46]. In another study in Nigeria, out of 32 smear-positive samples identified as not containing TB bacilli, six were NTM (18.75%), 14 were corynebacteria (43.75%), and 12 were rhodococci (37.5%) [28]. Our study did not further identify the non-MTBC/non-NTM samples. In addition, the lower sensitivity of the conventional nested PCR may have hampered confirmation by sequencing in case of negative results or low amplicon yield. Indeed, the in-house qPCR has already shown good sensitivity, while limited loss of DNA has been described during multiple handling steps for conventional nested PCR [47, 48]. This lower sensitivity is reflected in the sample's AFB grading, with fewer non-identified samples among the strong positive specimens.

This study was part of the national TB drug resistance survey, comprising new and previously treated cases. The concern of false-positive MTBC detection in sputum from previously treated patients has been chiefly raised for the highly sensitive, newer generation of rapid molecular assays, such as the "MTB detected Trace" result by Xpert Ultra [49, 50]. Nevertheless, even smear microscopy can still show non-viable AFB and the introduction of alternative/additional diagnostic assays or flows would need to be validated against culture.

Our survey was not designed to include culture in the diagnostic algorithm, and hence, we could not assess the viability of the detected MTBC and NTM bacilli, nor whether co-detection of MTBC and NTM reflected simultaneous infection, or rather an NTM infection following MTBC infection in previously treated patients, which was still detectable due to remnant DNA.

Also, we did not investigate whether the detected NTM was attributable to actual infection, colonization, or sample contamination. We could not assess the American Thoracic Society criteria to detect NTM in at least 3 different samples per patient, given the survey's design and

the retrospective nature of molecular analysis to detect NTM. However, study participants were people with respiratory signs undergoing evaluation for pulmonary TB attending health centers or patients failing TB treatment. Finally, we do not describe the outcome of patients in whom NTM was detected. Still, all were started on 1st line TB therapy or multidrug-resistant treatment following the results of initial microscopy and Xpert testing.

## 5. Conclusion

Among sputum-AFB-positive patients presenting in Mali, 3.4% were due to NTM. Particular attention must be paid to populations at risk, such as elderly patients and patients with a history of previous treatment. TB prevention and care programmes in resource-limited countries should ensure that their routine diagnostic algorithms can detect such cases, based on the concurrent use of rapid molecular TB diagnostics and microscopy without other molecular assays to detect NTM. However, culture is still necessary to isolate these mycobacteria and to monitor treatment in the event of confirmation of NTM disease. Clinical assessment and management of NTM infections remain a challenge worldwide.

## Supporting information

**S1 File. Minimal data set.** ID = Identification; HIV = Human Immunodeficiency Virus; MTB = *Mycobacterium tuberculosis* complex; qPCR = quantitative Polymerase Chain Reaction; rRNA PCR = ribosomal Ribonucleid Acid Polymerase Chain Reaction; Ct = Cycle threshold. (CSV)

## Acknowledgments

We would like to thank the TB management sites for recruiting patients, collecting and transporting samples, the country's regional health departments for regional coordination of the drug resistance study, and Olga Tosas AUGUET for her assistance in planning the survey, monitoring the field activities and preliminary data analysis.

## Author Contributions

**Conceptualization:** Aissata Boubakar Cisse, Anna S. Dean, Armand Van Deun, Ibrehima Guindo.

**Data curation:** Aissata Boubakar Cisse, Anna S. Dean, Seydou Arama, Bouke de Jong, Leen Rigouts.

**Formal analysis:** Aissata Boubakar Cisse, Anna S. Dean, Willem-Bram De Rijk, Bouke de Jong, Leen Rigouts.

**Funding acquisition:** Aissata Boubakar Cisse.

**Investigation:** Aissata Boubakar Cisse.

**Methodology:** Aissata Boubakar Cisse, Anna S. Dean, Armand Van Deun, Jelle Keysers, Mourad Gumusboga, Hawa Samake, Seydou Arama, Bassirou Diarra, Ibrahim Djilla, Fatoumata N. Coulibaly, Hawa Simpara, Mamadou Berthe, Ibrehima Guindo, Bouke de Jong.

**Project administration:** Aissata Boubakar Cisse.

**Software:** Aissata Boubakar Cisse.

**Supervision:** Aissata Boubakar Cisse, Anna S. Dean.

**Validation:** Aissata Boubakar Cisse, Anna S. Dean.

**Visualization:** Aissata Boubakar Cisse, Khadidia Ouattara, Yacouba Toloba.

**Writing – original draft:** Aissata Boubakar Cisse, Anna S. Dean, Armand Van Deun.

**Writing – review & editing:** Aissata Boubakar Cisse, Anna S. Dean, Armand Van Deun, Ibrehima Guindo, Bouke de Jong, Leen Rigouts.

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
