## [Decision Letter · Decision Letter 0]

14 May 2024

PONE-D-24-10019Prevalence of non-tuberculous mycobacteria among people presumed to have tuberculosis, positive for acid-fast bacilli in MaliPLOS ONE

Dear Dr. CISSE,

Thank you for submitting your manuscript to PLOS ONE. After careful consideration, we feel that it has merit but does not fully meet PLOS ONE’s publication criteria as it currently stands. Therefore, we invite you to submit a revised version of the manuscript that addresses the points raised during the review process.

The reviewers have recommended publication, but also suggest significant revisions to your manuscript.  Therefore, I invite you to respond to the reviewers' comments and revise your manuscript.

We look forward to receiving your revised manuscript.

Kind regards,

Fumihiro Yamaguchi

Academic Editor

PLOS ONE

https://journals.plos.org/plosone/s/file?id=ba62/PLOSOne_formatting_sample_title_authors_affiliations.pdf"

2.Please provide additional details regarding participant consent. In the ethics statement in the Methods and online submission information, please ensure that you have specified what type you obtained (for instance, written or verbal, and if verbal, how it was documented and witnessed). If your study included minors, state whether you obtained consent from parents or guardians. If the need for consent was waived by the ethics committee, please include this information.

Reviewers' comments:

Reviewer's Responses to Questions

**Comments to the Author**

1. Is the manuscript technically sound, and do the data support the conclusions?

Reviewer #1: Partly

Reviewer #2: Yes

2. Has the statistical analysis been performed appropriately and rigorously? 

Reviewer #1: Yes

Reviewer #2: Yes

3. Have the authors made all data underlying the findings in their manuscript fully available?

Reviewer #1: Yes

Reviewer #2: Yes

4. Is the manuscript presented in an intelligible fashion and written in standard English?

Reviewer #1: Yes

Reviewer #2: No

5. Review Comments to the Author

Reviewer #1: Introduction part is not enough and Rewrite the introduction part with the current information

Major revision is required to get published

rewrite the contents in the methodology and discussion part as mentioned in the reviewed manuscript

Reviewer #2: Your article raises several questions about TB diagnosis and provides additional evidence for why smear microscopy as a first line diagnostic tool needed to be replaced with GeneXpert. However it is interesting that GeneXpert also missed some cases that your inhouse method detected. Could you explain how your method achieved this?

Title : Consider changing the title to 'Prevalence of non-tuberculous mycobacteria among acid-fast positive presumptive TB cases in Mali"

Methods: NTM is increasingly becoming an issue especially in the context of DR-TB. However it would be good if you can give more context to the study-The study was carried out in 2019 as a TB drug resistance study, was the NTM aspect done as part of the DR-TB study or archived Xpert-negative samples were retrieved and used for this study?

Based on your results, your in-house qPCR for IS6110 was more sensitive than Xpert as it detected some additional MTBC cases from Xpert negative samples? how sure are you that these additional cases represented ongoin infection with MTBC and not previous exposure? During the survey was TB culture (gold standard for TB diagnosis) done for the ZN positive samples to confirm ongoing infection?

During the survey, at which point were people treated for TB, after ZN staining or based on Xpert results? If based on Xpert results, what happened to those who were Xpert-negative and later MTBC was detected by your inhouse method?

Conclusion: The conclusion that there is the need for alternative diagnostics beyond Xpert is valid, as your study found that some Xpert-negative and AFB positive samples could actually still be MTBC positive by other methods. However it would be important to state that such methods would need to be validated with TB culture which is the gold standard.

General comments

Line 154-157: The sentence is too long and confusing, kindly rephrase using shorter sentences for easy comprehension.

Line 184: change from "male patients were more numerous to "male patients were more".

Line 214-216: "The proportion of NTM versus Nocardia spp note the clinical relevance of these isolated bacteria were discussed in that study". What is the relevance of the statement? Can you rephrase for clarity?

Line 252: Please check the arrangements of the references-some have been repeated and are outside the bracket.

Check spacing before references in brackets- In some instances, there is no space between the last word and the reference number. Please be consistent with this throughout the manuscript.

6. PLOS authors have the option to publish the peer review history of their article (what does this mean?). If published, this will include your full peer review and any attached files.

Reviewer #1: **Yes: **Muthuraj Muthaiah

Reviewer #2: No

---

## [Author Response · Author response to Decision Letter 0]

27 Jun 2024

The drug resistance survey protocol in which a secondary analysis of samples is described in the informed consent was accepted by the ethics committee of the National Institute of Public Health Research by decision No. 08/2018/CE-INRSP on March 21, 2018. The objectives and expected results of the study were clearly explained in the language understood by the patient. Patients were asked for permission to use their samples for additional analyses, including outside Mali. Oral and written consent were obtained from patients over 18 years of age. For patients under the age of 18, after obtaining the prior oral and written consent of the parent or companion, the child's assent to participate in the study was obtained in the presence of a witness other than the parent or companion.

---

## [Decision Letter · Decision Letter 1]

15 Jul 2024

PONE-D-24-10019R1Prevalence of non-tuberculous mycobacteria among people with acid-fast positive presumptive tuberculosis in MaliPLOS ONE

Dear Dr. CISSE,

Thank you for submitting your manuscript to PLOS ONE. After careful consideration, we feel that it has merit but does not fully meet PLOS ONE’s publication criteria as it currently stands. Therefore, we invite you to submit a revised version of the manuscript that addresses the points raised during the review process.

The reviewers have recommended publication, but also suggest significant revisions to your manuscript.  Therefore, I invite you to respond to the reviewers' comments and revise your manuscript.

We look forward to receiving your revised manuscript.

Kind regards,

Fumihiro Yamaguchi

Academic Editor

PLOS ONE

Journal Requirements:

Reviewers' comments:

Reviewer's Responses to Questions

**Comments to the Author**

1. If the authors have adequately addressed your comments raised in a previous round of review and you feel that this manuscript is now acceptable for publication, you may indicate that here to bypass the “Comments to the Author” section, enter your conflict of interest statement in the “Confidential to Editor” section, and submit your "Accept" recommendation.

Reviewer #1: All comments have been addressed

Reviewer #2: All comments have been addressed

2. Is the manuscript technically sound, and do the data support the conclusions?

Reviewer #1: Yes

Reviewer #2: Yes

3. Has the statistical analysis been performed appropriately and rigorously? 

Reviewer #1: Yes

Reviewer #2: Yes

4. Have the authors made all data underlying the findings in their manuscript fully available?

Reviewer #1: Yes

Reviewer #2: Yes

5. Is the manuscript presented in an intelligible fashion and written in standard English?

Reviewer #1: Yes

Reviewer #2: Yes

6. Review Comments to the Author

Reviewer #1: Manuscript focused mainly on NTM. If author had used few reference stain(ATCC), manuscript would have completed scientifically. Otherwise the manuscript required minor revision

Reviewer #2: Comments raised have been addressed adequately by the authors. Spaces before in text references have been addressed. The use of archived samples and the need for consent has also been explained. The conclusion has been expanded to include that culture still remains tee Gold standard for TB diagnosis and should be considered as such even in the case of NTM

7. PLOS authors have the option to publish the peer review history of their article (what does this mean?). If published, this will include your full peer review and any attached files.

Reviewer #1: **Yes: **Muthuraj Muthaiah

Reviewer #2: No

---

## [Author Response · Author response to Decision Letter 1]

8 Aug 2024

Journal Requirements

We have reviewed our list of references and have not made any changes.

Point raised by Reviewer #1 

We have added some reference strains used for our sequence analysis. We described it in the manuscript on lines 153-157: “Sequencing was subcontracted to BaseClear (BE Leiden, The Netherlands). After entering the sequence into the Basic Local Alignment Search Tool (BLAST) of the National Centre for Biotechnology Information (NCBI), we compared it to reference strains sequences on NCBI such as ATCC 25791, ATCC BAA-614, ATCC 27282, ATCC 51457, 4773, ATCC 700504, ATCC 25276, ATCC 35752, ATCC 13950”.

---

## [Decision Letter · Decision Letter 2]

1 Sep 2024

PONE-D-24-10019R2Prevalence of non-tuberculous mycobacteria among people with acid-fast positive presumptive tuberculosis in MaliPLOS ONE

Dear Dr. CISSE,

Thank you for submitting your manuscript to PLOS ONE. After careful consideration, we feel that it has merit but does not fully meet PLOS ONE’s publication criteria as it currently stands. Therefore, we invite you to submit a revised version of the manuscript that addresses the points raised during the review process.

The reviewers have recommended publication, but also suggest significant revisions to your manuscript.  Therefore, I invite you to respond to the reviewers' comments and revise your manuscript.

We look forward to receiving your revised manuscript.

Kind regards,

Fumihiro Yamaguchi

Academic Editor

PLOS ONE

Journal Requirements:

Reviewers' comments:

Reviewer's Responses to Questions

**Comments to the Author**

1. If the authors have adequately addressed your comments raised in a previous round of review and you feel that this manuscript is now acceptable for publication, you may indicate that here to bypass the “Comments to the Author” section, enter your conflict of interest statement in the “Confidential to Editor” section, and submit your "Accept" recommendation.

Reviewer #1: All comments have been addressed

Reviewer #2: All comments have been addressed

2. Is the manuscript technically sound, and do the data support the conclusions?

Reviewer #1: Yes

Reviewer #2: Yes

3. Has the statistical analysis been performed appropriately and rigorously? 

Reviewer #1: Yes

Reviewer #2: Yes

4. Have the authors made all data underlying the findings in their manuscript fully available?

Reviewer #1: Yes

Reviewer #2: Yes

5. Is the manuscript presented in an intelligible fashion and written in standard English?

Reviewer #1: Yes

Reviewer #2: Yes

6. Review Comments to the Author

Reviewer #1: The revised manuscript contains minor corrections

Author should address the corrections before get published

Reviewer #2: Thank you for addressing to all the comments satisfactorily especially with regards to attaching a version of the the manuscript indicating with track changes, the revisions the had been made. I have no further comments at this stage .

7. PLOS authors have the option to publish the peer review history of their article (what does this mean?). If published, this will include your full peer review and any attached files.

Reviewer #1: **Yes: **Muthuraj Muthaiah

Reviewer #2: No

---

## [Author Response · Author response to Decision Letter 2]

16 Sep 2024

We have reviewed our list of references and have not made any changes. And we have accepted the minor corrections.

---

## [Decision Letter · Decision Letter 3]

2 Oct 2024

PONE-D-24-10019R3Prevalence of non-tuberculous mycobacteria among people with acid-fast positive presumptive tuberculosis in MaliPLOS ONE

Dear Dr. CISSE,

Thank you for submitting your manuscript to PLOS ONE. After careful consideration, we feel that it has merit but does not fully meet PLOS ONE’s publication criteria as it currently stands. Therefore, we invite you to submit a revised version of the manuscript that addresses the points raised during the review process.

**The reviewers have recommended publication, but also suggest significant revisions to your manuscript.  Therefore, I invite you to respond to the reviewers' comments and revise your manuscript.**

We look forward to receiving your revised manuscript.

Kind regards,

Fumihiro Yamaguchi

Academic Editor

PLOS ONE

**Journal Requirements:**

Reviewers' comments:

Reviewer's Responses to Questions

**Comments to the Author**

1. If the authors have adequately addressed your comments raised in a previous round of review and you feel that this manuscript is now acceptable for publication, you may indicate that here to bypass the “Comments to the Author” section, enter your conflict of interest statement in the “Confidential to Editor” section, and submit your "Accept" recommendation.

Reviewer #1: All comments have been addressed

2. Is the manuscript technically sound, and do the data support the conclusions?

Reviewer #1: Partly

3. Has the statistical analysis been performed appropriately and rigorously? 

Reviewer #1: Yes

4. Have the authors made all data underlying the findings in their manuscript fully available?

Reviewer #1: Yes

5. Is the manuscript presented in an intelligible fashion and written in standard English?

Reviewer #1: Yes

6. Review Comments to the Author

**Reviewer #1: **This manuscript require minor revision before to get published as per the attachment of the revised manuscript

7. PLOS authors have the option to publish the peer review history of their article (what does this mean?). If published, this will include your full peer review and any attached files.

Reviewer #1: **Yes: **Muthuraj Muthaiah

---

## [Author Response · Author response to Decision Letter 3]

22 Oct 2024

We have reviewed our list of references and have not made any changes.

Points raised by Reviewer #1: This manuscript require minor revision before to get published as per the attachment of the revised manuscript.

We have accepted the minor corrections. But we proposed for sentence 205-206: “The microscopy laboratories provided AFB quantification result for 1414 of the 1418 patients” instead of “The participating laboratories were informed of the quantification of AFB for 1414 out of 1418 patients”.

---

## [Decision Letter · Decision Letter 4]

28 Nov 2024

Prevalence of non-tuberculous mycobacteria among people with acid-fast positive presumptive tuberculosis in Mali

PONE-D-24-10019R4

Dear Dr. CISSE,

We’re pleased to inform you that your manuscript has been judged scientifically suitable for publication and will be formally accepted for publication once it meets all outstanding technical requirements.

Kind regards,

Fumihiro Yamaguchi

Academic Editor

PLOS ONE

Additional Editor Comments (optional):

This paper has adequately addressed the reviewer's comments, and no further revisions are deemed necessary.

Reviewers' comments:

Reviewer's Responses to Questions

**Comments to the Author**

1. If the authors have adequately addressed your comments raised in a previous round of review and you feel that this manuscript is now acceptable for publication, you may indicate that here to bypass the “Comments to the Author” section, enter your conflict of interest statement in the “Confidential to Editor” section, and submit your "Accept" recommendation.

Reviewer #1: All comments have been addressed

2. Is the manuscript technically sound, and do the data support the conclusions?

Reviewer #1: Yes

3. Has the statistical analysis been performed appropriately and rigorously? 

Reviewer #1: Yes

4. Have the authors made all data underlying the findings in their manuscript fully available?

Reviewer #1: Yes

5. Is the manuscript presented in an intelligible fashion and written in standard English?

Reviewer #1: Yes

6. Review Comments to the Author

Reviewer #1: Author addressed all queries of reviewer. Even though the manuscript requires Minor revision before to get published

7. PLOS authors have the option to publish the peer review history of their article (what does this mean?). If published, this will include your full peer review and any attached files.

Reviewer #1: **Yes: **Muthuraj Muthaiah

---

## [Editor Report · Acceptance letter]

4 Dec 2024

PONE-D-24-10019R4 

PLOS ONE

Dear Dr. CISSE, 

I'm pleased to inform you that your manuscript has been deemed suitable for publication in PLOS ONE. Congratulations! Your manuscript is now being handed over to our production team.

Kind regards, 

on behalf of

Dr. Fumihiro Yamaguchi 

Academic Editor

PLOS ONE